# Fast room temperature lability of aluminosilicate zeolites

Christopher J. Heard [1], Lukas Grajciar [1], Cameron M. Rice[2], Suzi M. Pugh[2], Petr Nachtigall [1]*, Sharon E. Ashbrook [2]* & Russell E. Morris [1,2]*

Aluminosilicate zeolites are traditionally used in high-temperature applications at low water vapour pressures where the zeolite framework is generally considered to be stable and static. Increasingly, zeolites are being considered for applications under milder aqueous conditions. However, it has not yet been established how neutral liquid water at mild conditions affects the stability of the zeolite framework. Here, we show that covalent bonds in the zeolite chabazite (CHA) are labile when in contact with neutral liquid water, which leads to partial but fully reversible hydrolysis without framework degradation. We present ab initio calculations that predict novel, energetically viable reaction mechanisms by which Al-O and Si-O bonds rapidly and reversibly break at 300 K. By means of solid-state NMR, we confirm this prediction, demonstrating that isotopic substitution of $^{17}O$ in the zeolitic framework occurs at room temperature in less than one hour of contact with enriched water.

[1] Department of Physical and Macromolecular Chemistry, Faculty of Science, Charles University, Hlavova 8, Prague 2, Czech Republic. [2] School of Chemistry, EaStCHEM and Centre of Magnetic Resonance, University of St Andrews, Purdie Building, St Andrews, UK. *email: petr.nachtigall@natur.cuni.cz; sema@st-andrews.ac.uk; rem1@st-andrews.ac.uk

Zeolites are some of the most important materials in use today, with high volume applications ranging from ion exchange to heterogeneous catalysis, and emerging applications in energy and medicine[1]. As they are porous materials, zeolites are generally metastable when calcined (i.e. when any solvent or guest molecules are removed from the pores). However, a particularly valued feature of aluminosilicate zeolites, the type that is of most importance in industry, is their perceived stability under mild conditions. This stability is pivotal for many applications and is particularly important when nucleophiles such as water are present. The presence of relatively strong Si–O and Al–O bonds in the materials, in addition to the wealth of average structure data from crystallographic studies, leads to the perception that zeolites are static compounds with pore windows that control access to the interior of the solids.

For many years there have been studies on how zeolites interact with vapour phase moisture, as 'steaming' is a common method of pretreatment in catalytic processes where aluminium is removed from the framework[2–5]. Isotopic enrichment of the framework via oxygen exchange with water vapour has been employed in several zeolites at elevated temperatures (above 200 °C)[6–8]. Such experiments have shown that exchange preferentially occurs at framework oxygen sites directly connected to aluminium. At high temperature, the water vapour pressure is low, limiting the exchange to mechanisms involving single or few water molecules[9–11]. However, under low temperature conditions, the water loading in the zeolite pores is much higher. This raises the possibility of novel collective mechanisms for framework breaking and oxygen exchange[12].

Recently, there has been great interest in how zeolites interact with liquid water at relatively modest temperatures around 100 °C (often known as hot liquid water)[13–17]. This interest is primarily driven by two different recent developments: first, the use of biomass as a feedstock for the replacement of fossil fuel refineries, where catalytic reactions often take place in water; and second, the use of water to hydrolyse zeolites as a route for the synthesis of novel zeolite structures that cannot be targeted using traditional preparative methods (the so-called ADOR method)[18–20]. In both cases the stability of zeolites when challenged with liquid water is key. In catalytic biomass refining, hot liquid water has been shown to degrade some zeolites, with the extent of degradation correlating with the number of defects in the material[20]. In the ADOR process, the controlled degradation of germanosilicate zeolites using water removes germanium selectively from the materials, forming intermediates that can then be further manipulated[19,21]. Both of these processes involve the presence of specific weaknesses in the zeolite framework – the presence of defects or a hydrolytically amenable substituent element such as germanium. In our previous studies on the mechanism of the ADOR process we noted that in liquid $^{17}$O-enriched water at just below 100 °C there was incorporation of the isotope not only into the interlayer region (as would be expected) but also into the bulk Si–O–Si units[22]. This result raised the question of whether $^{17}$O incorporation is achievable under similar conditions for other zeolites of catalytic importance. Furthermore, it is important to determine whether $^{17}$O incorporation is kinetically viable at even lower temperatures. $^{17}$O NMR is an excellent tool for following oxygen exchange reactions, and has been used in both solution-[23] and solid-state[22] experiments.

Here we report the results of a combined theoretical and experimental study of framework lability on the zeolite chabazite. Biased ab initio molecular dynamics (AIMD) studies predict, through the elucidation of a new reaction mechanism, that Al–O and Si–O bond-breaking processes should be possible even at ambient temperature. We test this prediction experimentally by 2D solid-state NMR techniques, showing that $^{17}$O incorporation into the framework from isotopically enriched water is very fast, with a signal from both Si–$^{17}$O–Si and Si–$^{17}$O–Al being visible within even a few minutes. It is therefore found that even under mild, neutral conditions, water is an active species, driving partial, reversible hydrolysis of the framework.

## Results

**Computational studies.** Zeolites with the chabazite (CHA) structure type are an excellent choice for computational studies as they have a relatively simple crystallographic structure. They are also important industrial materials, with applications in fields such as automotive emission catalysis[24] and methane partial oxidation[25,26]. The CHA framework in the form of SAPO-34 has been interrogated with molecular dynamics simulations for applications in methanol conversion, where the interaction of incorporated water and reactant molecules is important[27,28]. CHA has a high symmetry structure with only a single crystallographically independent tetrahedral site and four independent framework oxygen positions (Fig. 1). In this work, we have investigated the initial hydrolysis of zeolite CHA in liquid water with biased AIMD. The zeolite channels were fully loaded with water molecules (15 $H_2O$ molecules per 36 T site supercell, corresponding to a density of 1 g/mL within the microporous volume). Calculations predict that under such hydration conditions, both Al–O and Si–O bonds are susceptible to hydrolysis even at room temperature.

A novel reaction mechanism was found for Si–O bond hydrolysis, with free energy barriers, $\Delta A^{\neq}$, as calculated through thermodynamic integration of constrained ab initio molecular dynamics simulations, as low as 63 kJ mol$^{-1}$. This reaction mechanism, denoted axial, starts with interaction of a water molecule directly to silicon through its oxygen atom, resulting in a five-coordinate Si (Fig. 1a). The formation of a chain of solvent water molecules between this interacting water and the adjacent framework oxygen in an axial position is essential for the reaction to proceed. A proton is shuttled, in a Grotthuss-type mechanism through the water chain to the axial framework oxygen, which breaks the Si–O bond, inverting the $SiO_3OH$ tetrahedron, and forming two silanol groups that are in anti-positions to each other[29]. Consequently, such products are only a few kJ mol$^{-1}$ higher than reactants on the free energy surface. Thus, the reaction is both thermodynamically and kinetically feasible, in sharp contrast to the equatorial mechanism proposed in the literature, which gives barriers for Si–O–Si hydrolysis between 160–200 kJmol$^{-1}$[30–32]. (The full axial reaction path is provided in the SI in the form of movies.) The axial reaction mechanism has been observed for hydrolysis at O1 and O4 framework oxygen atoms (free energy barriers of 63 kJ mol$^{-1}$) and both reaction pathways are only slightly endergonic (1 and 24 kJ mol$^{-1}$, respectively). A similar mechanism is expected for the hydrolysis of the O3 oxygen atom, as O1, O2, and O3 are all part of the double six ring units present in the structure and have similar surroundings (Fig. 1). However, because of the framework topology we do not predict that the same mechanism will work for the O2 oxygen atom as access to the initial water adsorption would require water inside the double six ring unit and so would be very hindered. The small differences in reaction free energies are due to steric constraints of the reaction products. It is important to note that the axial mechanism is only available in the presence of sufficient solvent water to form chains. Involvement of the proton shuttle via the Grotthuss mechanism is entropically demanding and its probability will decrease with increasing temperature. The free energies of activation (as well as reaction free energies) account for this entropy contribution and AIMD simulations carried out at 450 K indeed show that the free

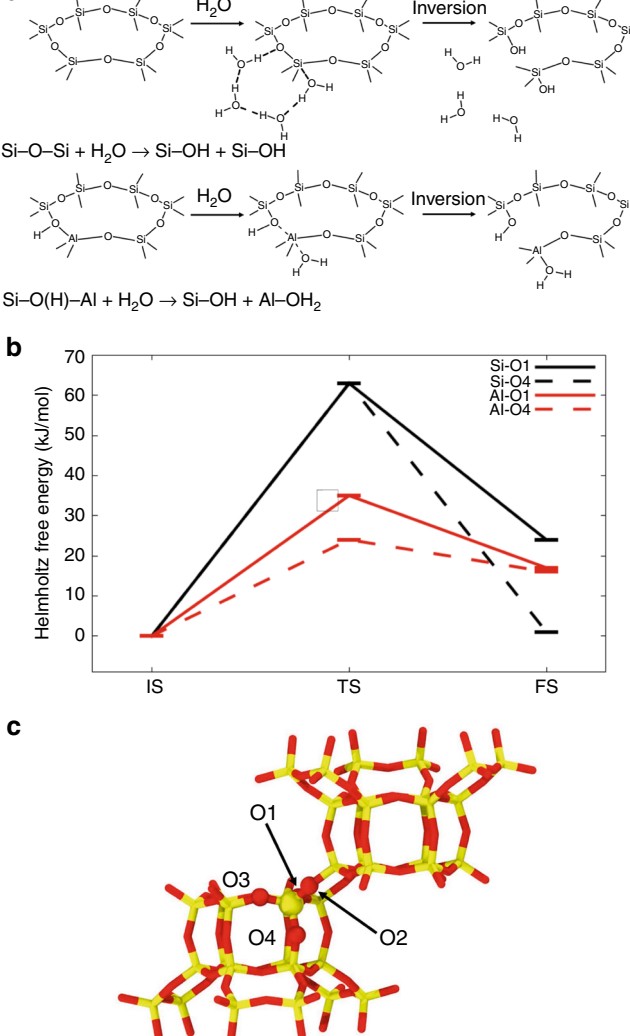

**Fig. 1** Calculated mechanism and energetics for the hydrolysis of bonds in zeolites. **a** The reaction mechanism for Si–O bond-breaking proceeding via non-dissociative water adsorption on Si, proton transfer via the Grotthuss mechanism, and Si–O bond breaking in an anti-position to adsorbed water, with inversion of the SiO$_3$OH tetrahedron. The Al–O bond breaking occurs simultaneously with the non-dissociative water adsorption on Al in the anti-position to a Bronsted acid site followed by inversion of the AlO$_3$.H$_2$O tetrahedron. **b** The relative Helmholtz free energies of initial compounds (IS), transition states (TS) and final products (FS) of the hydrolysis of Al–O and Si–O at O1 and O4 sites. **c** Schematic of the Si-only CHA cell showing the four crystallographically distinct oxygens atoms: yellow indicates the silicon atoms and red the oxygens

energies of activation increase (see Supplementary Information Table 2).

The dimensions of zeolite channels and cages are too small for formation of bulk-like water structure. The structure of the liquid water at sufficient density, such as is found inside the zeolite pore, favours this mechanism. Analysis of the hydrogen bonding network of the water during unbiased equilibration dynamics simulations reveals an average number of 0.86 H-bonds per water oxygen, which is lower than the value of 2 to be expected in unconstrained bulk water at the same density, and close to the value of 1 expected for a perfect linear chain. Such a complex water network is due primarily to the topology of the zeolite void space, which comprises small cages connected by narrow (8-ring) windows. Additionally, the water is protonated to a degree

estimated at ~99%[33], due to the equilibrium between Brønsted proton adsorption at the Al–O–Si group, and solvation in the water[34–36]. At 300 K, we find this equilibrium to lie on the side of solvation (See Supplementary Figs. 5 and 6).

Even lower free energy barriers were found for hydrolysis of the Al–O bond. Due to the presence of a Brønsted proton, the mechanism is simpler than for Si–O scission. The mechanism involves the adsorption of water through an oxygen atom to framework Al, followed by the inversion of the AlO$_4$ tetrahedron. A concurrent breakage of the Al–O bond leads to the intermediate hydrolysis product. This mechanism is rather similar to the one proposed by Silaghi et al. for Al–O bond hydrolysis by a single water molecule[10,37], though in the present calculations the proton is adsorbed from the solvated state during the reaction.

Calculations thus reveal an unexpected lability of zeolite framework; both Si–O and Al–O bonds can break under wet conditions at room temperature. While faster dynamics are predicted for the Al–O bond, the hydrolysis products of Si–O–Si bond breaking are thermodynamically slightly more stable. We predict that: (i) Under wet conditions a dynamic equilibrium is established between the hydrolysed and un-hydrolysed zeolite framework. (ii) This equilibrium is on the side of the un-hydrolyzed structure of Si–O–Al and Si–O–Si bonds. (iii) It is thermodynamically more favourable to form Q$^3$ silanols from Si–O–Si than Si–O–Al, but that Si–O–Al bonds will hydrolyse slightly faster. (iv) Low-temperature framework lability, which is a necessary first step towards isotopic exchange of oxygen, is likely under wet conditions. These results predict that a softer zeolite framework and spontaneous ring opening occurs due to the presence of water. This will lead to changes in mass transport and even to changes in reaction selectivity. The presence of transient hydroxyl groups will also affect the interaction energies between guest molecules and the zeolite internal surface, impacting on diffusion, and therefore reaction kinetics.

**Solid-state NMR studies.** The lability of zeolite linkages within the CHA framework can be demonstrated experimentally using multinuclear (i.e. $^1$H, $^{17}$O, $^{27}$Al, and $^{29}$Si) solid-state NMR spectroscopy. NMR spectra were taken for a 'slurry' consisting of 1:1 µL:mg ratio of 40% enriched H$_2$$^{17}$O (l) and SSZ-13 zeolite (CHA framework, Si/Al = 11) over a period of 250 days. $^{17}$O MAS (magic angle spinning) NMR spectra of the zeolite slurries confirm room temperature $^{17}$O enrichment of the framework on a rapid timescale. Signal is observed in an MAS spectrum taken only ~1 h after exposure (Fig. 2a). Although the sharp signal at ~0 ppm results from H$_2$$^{17}$O (l), the broader signals at higher shift are attributed to the framework O species. This can be confirmed by a 3QMAS spectrum (Fig. 2b) which resolves signals for Si–O–Si ($\delta_1 \approx 33$ ppm) and Si–O–Al ($\delta_1 \approx 20$–25 ppm) linkages, confirming rapid enrichment of both types of framework O sites. The NMR parameters (Si–O–Si P$_Q \approx 5.3$–5.7 MHz, $\delta_{iso} \approx 44$ ppm; Si–O–Al P$_Q \approx 3.6$–4.0 MHz, $\delta_{iso} \approx 30$–37 ppm) are in good agreement with similar O species in other zeolites in the literature[7,38–40]. At higher field ($B_0 = 20.0$ T, Supplementary Fig. 2 and Supplementary Table 1 in the ESI), the signal attributed to Si–O–Al species is clearly split into two different resonances, confirming that two different types of Si–O–Al linkages are present. The signal at ~0 ppm (marked *) results from break-through of H$_2$$^{17}$O (l). $^{17}$O MAS and MQMAS spectra of a slurry aged for a much longer time (28 days) are shown in Fig. 2c, d, respectively. The intensity of the signals from the framework O species in the MAS spectrum has increased relative to that of H$_2$$^{17}$O (l), and signals from Si–O–Si and Si–O–Al species are both still present in the MQMAS spectrum. Supplementary Fig. 1 in

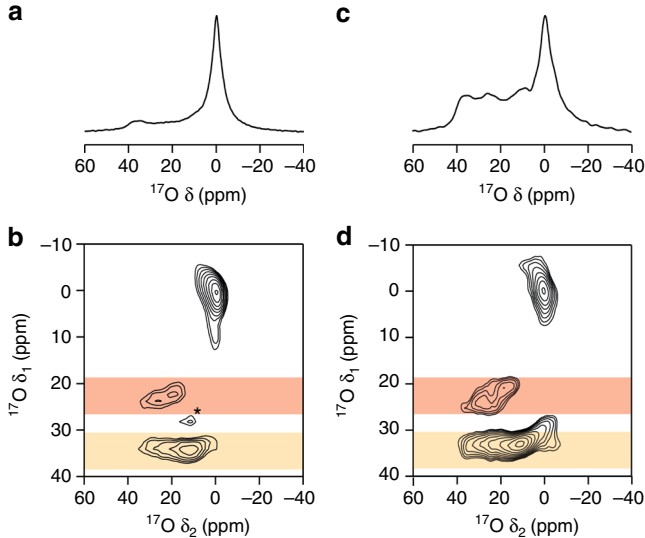

**Fig. 2** Magic-angle spinning (MAS) triple-quantum (3QMAS) NMR $^{17}O$ spectra for the zeolite treated with water. $^{17}O$ (14.1 T, 10 kHz) **a**, **c** MAS and **b**, **d** $^{1}H$ decoupled 3QMAS NMR spectra of a slurry of 40% $H_2^{17}O$ and Al-CHA aged for **a**, **b** ~1 h and **c**, **d** ~ 28 days at room temperature. Regions highlighted in red are attributed to Si–O–Al species and and those in gold attributed to Si–O–Si. The signal marked with asterisk (*) is attributed to water. The acquisition began **a** 10 mins and **b** 1 h, respectively, following the combination of enriched water and zeolite

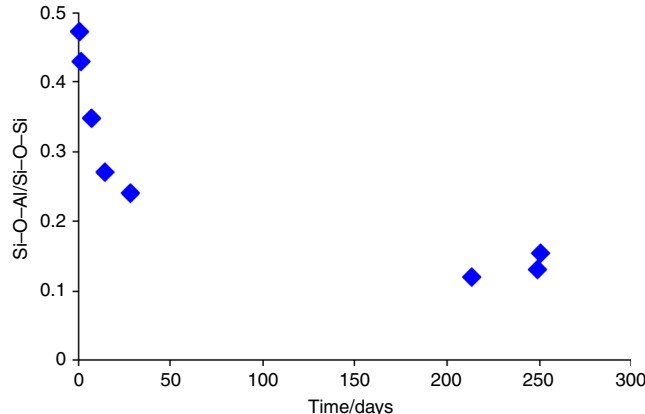

**Fig. 3** The evolution of relative $^{17}O$ signals over time. Plot of relative integrated intensities (Si–O–Al/Si–O–Si) from $^{17}O$ 3QMAS NMR spectra of Al-CHA/$H_2^{17}O$ slurries as a function of slurrying duration

the ESI shows that there is a consistent decrease in $^{17}O$ signal attributed to free $H_2^{17}O$ (l) as the slurrying time increases, and this signal is almost removed completely at very long durations (i.e. over 200 days). In addition to exchange with the framework O species it is also likely that a some free $H_2O$ (l) is lost through slow leakage from the rotor during this time. This is supported by the $^{1}H$ MAS NMR spectra (Supplementary Fig. 3, ESI), where at much longer slurrying times (i.e. over 100 days) an increase in linewidth is observed, suggesting a reduction in dynamics and the loss of rapidly reorienting $H_2O$ (l).

Figure 2 shows that oxygen enrichment of the zeolite framework occurs rapidly at room temperature. Owing to the inherently non-quantitative nature of 3QMAS spectra it is difficult to determine the absolute amount of framework enrichment. Further, the need to integrate over all spinning sidebands within the spectrum makes it difficult to accurately calculate the enriched Si–O–Al/Si–O–Si species ratio. However, it is clear from Fig. 3, which plots the relative intensities (i.e. integrated intensity including all spinning sidebands) of the two signals (as the Si–O–Al/Si–O–Si ratio) as a function of slurrying time, that there is some preferential enrichment of Si–O–Al species at shorter durations. At longer times, the Si–O–Al/Si–O–Si ratio is ~ 0.15, which is in relatively good agreement with the theoretical ratio for the CHA framework (of 0.17, assuming Löwenstein's rule holds) if uniform (relative) enrichment was obtained.

The $^{17}O$ enrichment of Al-CHA is not accompanied by significant degradation of the zeolite framework. This can be seen in Fig. 4, where the $^{29}Si$ and $^{27}Al$ MAS NMR spectra of a sample after slurrying show no noticeable changes from the corresponding spectra of the starting zeolite. The $^{29}Si$ MAS NMR spectrum of the starting Al-CHA zeolite (Fig. 4a) contains two major resonances (at $\delta \approx -110$ and $-105$ ppm) that can be attributed to $Q^4$ Si species with 0 and 1 Al neighbour, respectively. Lower signal intensity is observed at higher $\delta$ ($-100$ ppm), which could result from the presence of $Q^4$ (2 Al) Si species. However, the $^{1}H$-

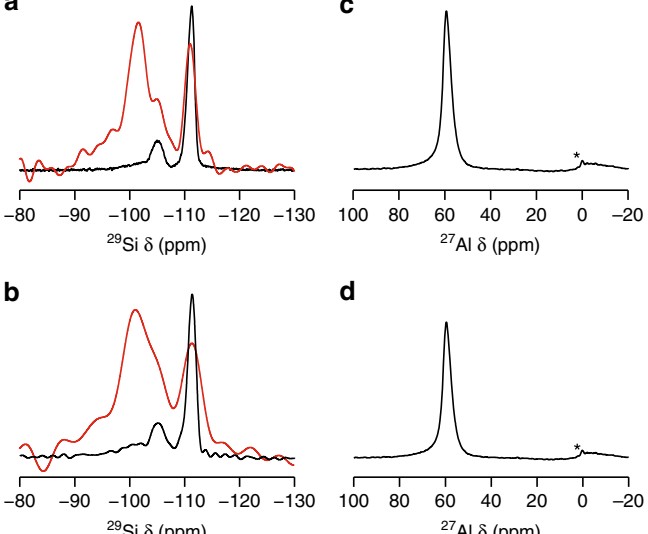

**Fig. 4** The evolution of $^{29}Si$ and $^{27}Al$ MAS and cross-polarisation (CP) MAS NMR spectra over time. $^{29}Si$ (**a** and **b**) and $^{27}Al$ (**c** and **d**) MAS NMR (9.4 T, 10 kHz) spectra of the starting Al-CHA zeolite (**a** and **c**) and a slurry of 40% $H_2^{17}O$ Al-CHA slurry aged for **b** 137 days (MAS), 45 days (CP MAS) and **d** 202 days. In **a** and **b** the corresponding $^{1}H/^{29}Si$ CP MAS NMR spectra (5000 μs contact time) are shown overlaid in red

$^{29}Si$ CP MAS NMR spectrum (where magnetisation is transferred from $^{1}H$ to nearby $^{29}Si$ species) overlaid in red in Fig. 4a, shows that some $Q^3$ Si species ($\delta \approx -102$ ppm) are present in the dry, un-slurried starting zeolite. It is therefore, difficult to decompose accurately the contribution of the $Q^4$ (2 Al) and $Q^3$ Si to the signal in this region. However, the spectra acquired after slurrying (Fig. 4b) are very similar with only a very small change in the intensity of the signal at this point, confirming no significant formation of additional $Q^3$ Si species, and that the framework is intact. The $^{27}Al$ MAS NMR spectra of the samples before (Fig. 4c) and after (Fig. 4d) exchange both show Al is present in tetrahedral coordination with no significant additional octahedral Al produced. The narrow line observed in both cases reflects the very small $^{27}Al$ quadrupolar coupling constant, consistent with the symmetric, tetrahedral nature of the framework. It should be noted that although no significant signal is observed in the $^{17}O$ NMR spectra for Si–OH or Al–OH species (consistent with the

lack of $Q^3$ Si species discussed above) it can be difficult to detect these signals in MQMAS spectra if dynamics of water are present, owing to rapid relaxation. This was shown clearly in recent work on $^{17}O$ NMR of Ge-UTL[22].

From the experimental data, we conclude: (i) Isotopic exchange of framework oxygen atoms is fast even at room temperature, it can be observed readily in slurries <24 h old. (ii) Isotopic enrichment of both Si–O–Si and Si–O–Al framework oxygen atoms is observed, with preferential enrichment of Si–O–Al species at shorter times. (iii) No significant degradation of the framework is observed, with only tetrahedral Al species seen, and no signficant increase in $Q^3$ Si species during slurrying. We also note that leaving the enriched samples exposed to wet air leads to no change in $^{17}O$ signal, similar to results previously reported by Stebbins and co workers[38]. This shows that liquid water is necessary for the enrichment to occur and at ambient temperature vapour phase water does not promote enrichment.

One fundamental question that needs to be addressed is the role of the defects in the enrichment process. It is not possible to accurately quantify the $Q^4/Q^3$ ratio in the initial zeolite owing to the potential overlap of signals from $Q^4(2\ Al)$ sites (from −100 to −101 ppm). Assuming a zeolite with no $Q^4(2\ Al)$, we can estimate the starting material to contain an absolute maximum of 5% $Q^3$ Si, but this is likely to be much lower owing to the presence of the $Q^4(2\ Al)$ signals. Given that the mechanism proposed in Fig. 1 essentially forms a transient $Q^3$ species we cannot distinguish experimentally between the the enrichment starting from $Q^3$ species or from attack at $Q^4$ species. It is likely that some initial enrichment takes place at the silanol defect species, but we can say for certain that all enrichment cannot be at the $Q^3$ sites as the relatively low level of 5% defects cannot account for the significant $^{17}O$ signal in the NMR studies.

The traditional view of zeolites is one of static materials, although it has been known for some while that they are flexible to some degree through breathing processes where no bonds are broken[41,42]. No previous work has ever demonstrated room temperature lability in zeolites. This is in contrast to metal-organic frameworks, another important family of nanoporous materials, where lability (and hemilability) are well established[43]. We have performed a combined computational and experimental study of hydrolysis in zeolite CHA at room temperature. Based on the results presented, we predict and observe a high degree of framework lability towards exchange of oxygen atoms in the framework. Such lability is present for both Si–O–Si and Al–O–Si moieties, and is likely to be observed in other aluminosilicate zeolites under wet conditions. The ease by which the framework is disrupted leads to facile oxygen isotope exchange, but which does not lead to framework degradation. In catalytic applications, new reaction mechanisms that take into account the presence of broken framework bonds become conceivable, while the understanding of processes that rely on low-temperature hydrolysis, such as targeted zeolite synthesis via the ADOR process, are also improved. This work is also important for our understanding of fundamental aspects of zeolites, such as molecular sieving and ion exchange. Fast bond-breaking and bond-making in water, even at room temperature, means that pore window sizes are not, as is commonly thought, the sole determining factor of what molecules can pass into the interior of the solid. Therefore, we propose to qualitatively change the traditional view of zeolites as static, solid materials, to a novel view under which zeolites should be understood as labile, dynamic, fluctional frameworks with the potential for rich chemistry under hydration.

## Methods

**Preparation of the zeolite/water slurry**. The zeolite slurry was prepared using a sample of calcined Al-SSZ-13 (CHA framework, Si/Al = 11), provided by Chevron

Corporation. 25 mg of calcined zeolite was mixed with 25 μL of 40% $H_2^{17}O$ (l) in the body of a standard Bruker 4.0 mm HRMAS rotor insert. The insert was sealed with a stopper and screw and left on the laboratory bench before being studying after the durations shown in the figures.

**NMR spectroscopy**. Solid-state NMR spectra were acquired using Bruker Avance III spectrometers equipped with 9.4, 14.1, or 20.0 T wide-bore magnets. The sample slurry was placed into an HRMAS insert and packed into a 4.0-mm ZrO2 rotor, and rotated at a rate of 10 kHz. Magic-angle spinning (MAS) NMR spectra were acquired at Larmor frequencies of 850 and 400 MHz (1 H), 104.23 MHz ($^{27}Al$), 79.47 MHz ($^{29}Si$) and 81.34 MHz, and 115.3 MHz for $^{17}O$, using a conventional 4.0 mm HX low-γ probe. Spectra are referenced to Si(CH$_3$)$_4$ for $^1H$ (using a secondary solid reference of l-alanine ($\delta$(NH$_3$) = 8.5 ppm)), 0.5 M Al(NO$_3$)$_3$ (aq) for $^{27}Al$ (using Al(acac)$_3$ ($\delta$COG = −4.2 ppm) as a secondary solid reference), and H$_2$O ($\delta$iso = 0.0 ppm) for $^{17}O$ at room temperature. $^{17}O$ MAS NMR spectra were recorded at 14.1T using two different excitation pulse lengths (0.5 and 1.0 μs) depending on the age of the sample. At 14.1T and 20.0T, 17O multiple quantum MAS (MQMAS) experiments were carried out using a triple-quantum z-filtered pulse sequence and are shown after a shearing transformation (using the convention described in ref. 3)[44–46].

**Simulation model**. Dynamical simulations were performed to investigate the first step of chabazite hydrolysis according to two possible reactions: Si-O cleavage in Si-only CHA and Al–O cleavage in aluminosilicate CHA. The 36T site supercell model for chabazite was adopted, which contains three cages. An Si:Al ratio of 11 was chosen for the aluminosilicate form. The structure of the zeolite was obtained from the IZA database, and locally reoptimized in the presence of water ($a$ = 13.76 Å, $b$ = 13.81 Å, $c$ = 14.30 Å, $\alpha$ = 90.05°, $\beta$ = 89.88°, $\gamma$ = 119.79°). Solvent water was included into the pore structure, with a density of ~1 g/cm$^3$, which corresponds to 15 molecules per supercell. The adequacy of employing such water loading in the aluminosilicate CHA model was verified by an experimental water adsorption isotherm for an H-CHA sample with Si/Al = 11 at 293 K (Supplementary Fig. 4), reporting an average of 12–15 water molecules per supercell. The CHA accessible volume fraction of 17.4% was taken from the IZA database. Molecules were initially placed in the zeolite pore structure by overlapping the unit cell with a cubic water box containing a random packing of spc water molecules with the gromacs solvate code[47], with a vdW radius scaling factor adjusted to match the number of molecules required to satisfy the required density.

**Simulation methods**. All calculations were performed using density functional theory as implemented in the VASP 5.4 code[48–51], with the generalised gradient exchange correlation functions of Perdew, Burke and Ernzerhof[52], and an additional dispersion correction, of the D3 form of Grimme et al. with Becke-Johnson damping[53,54]. Gaussian smearing of the Fermi-Dirac distribution was implemented, with a smearing width of 0.1 eV. Electronic states were sampled at the Gamma point only. Wavefunctions are described by plane-wave basis expanded to a kinetic energy cutoff of 400 eV. The stabilities of reactants and products were determined by ab initio molecular dynamics simulations of 10–15 ps with a timestep of 0.5 fs, employing the Nosé-Hoover thermostat in the NVT ensemble, with a target temperature of 300/450 K. Metadynamics simulations were used for the exploration of the free energy surface and generation of suitable collective variables. Configurations from the equilibration trajectories were selected and used as starting points for metadynamics simulations, with gaussian biasing potentials of height 0.05 eV and width 0.05 eV added every 50 fs (100 steps). Collective variables (CV) which describe the reaction step of interest were determined. The best one-dimensional collective variable, which smoothly varies from reactant to product with a low free energy barrier was chosen to represent the reaction. For the axial mechanism of Si–O cleavage, the CV is the norm of the vector consisting of the Si–O(water) distance, the H(water)-O(framework) distance, and the hydrogen bonds between the solvent waters in the chain, of which there are four. For Al–O (H) cleavage, the CV is the distance between aluminium and the oxygen of the incoming water molecule. These CVs are depicted in the Supplementary Fig. 5. The calculation of free energy barriers and pathways was performed as follows:

An estimate of the irreversible work along the reaction profile defined by the CV was obtained via the approximate slow growth method. The chosen CV was varied linearly along the reaction path via slow growth simulations of duration 20,000–25,000 steps. Constrained molecular dynamics, utilising blue moon sampling, in which the system samples the hypersurface defined by a fixed value of the CV, was employed to generate Helmholtz free energy gradients at a set of between 15 and 32 configurations selected to span the reaction coordinate between reactant and product states. Sampling was applied for 5 ps simulations (with a timestep of 0.5 ps) at each selected point along the trajectory. The final free energy profile was then generated through integration of the free energy gradient along the reaction path from reactant to product. This integration provides the final estimate of the minimum free energy path and thereby the estimate of the Helmholtz free energy barrier is obtained.

## Data availability

Data supporting the findings of this manuscript are available from the corresponding authors upon reasonable request. Electronic data underpinning this manuscript is available at https://doi.org/10.17630/31d75f26-ea55-42c9-9e06-c19bb4325200.

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

## Acknowledgements

Charles University Centre of Advanced Materials (CUCAM) (OP VVV Excellent Research Teams, project number CZ.02.1.01/0.0/0.0/15_003/0000417) is acknowledged.

P.N. acknowledges the Czech Science Foundation (19–21534 S). This work was supported by The Ministry of Education, Youth and Sports from the Large Infrastructures for Research, Experimental Development and Innovations project "IT4Innovations National Supercomputing Center – LM2015070". The UK 850 MHz solid-state NMR Facility used in this research was funded by EPSRC and BBSRC (contract reference PR140003) as well as the University of Warwick including via part funding through Birmingham Science City Advanced Materials Projects 1 and 2 supported by Advantage West Midlands (AWM) and the European Regional Development Fund (ERDF). Collaborative assistance from the 850 MHz Facility Manager (Dinu Iuga, University of Warwick) is acknowledged. This work was also supported by the ERC (EU FP7 Consolidator Grant 614290 "EXONMR" and Advanced Grant 787073 "ADOR") and the EPSRC (EP/N509759/1 and EP/N50936X/1). S.E.A. would like to thank the Royal Society and the Wolfson Foundation for a merit award. We would like to thank Dr Stacey Zones of Chevron for providing a sample of SSZ-13 with the CHA structure. We would also like to thank Qiudi Yue and Dr Maksym Opanasenko for providing the sample for water adsorption isotherm measurements, and Dr Martin Kubů for performing the experiment.

## Author contributions

P.N., S.E.A., and R.E.M. conceived the project and contributed to the design of the experiments and calculations. C.J.H. and L.G. performed the calculations and analysed the simulation results. C.M.R. and S.M.P. designed and performed the NMR spectroscopic measurements and analysed the experimental results. C.J.H., C.M.R., S.E.A and R.E.M. wrote the paper. All the authors discussed the results, commented on and edited the manuscript.

## Competing interests

The authors declare no competing interests.
