## [Peer Review File · Nature Communications]

Reviewers' comments:

Reviewer #1 (Remarks to the Author):

Probing the function of water in zeolite is an important and challenge issue, which is essential to understand stability and activity of zeolite. ^{17}O solid-state NMR coupling with ab initio calculations is a promising approach to explore how neutral liquid water at mild conditions affects the stability of the zeolite framework. In this manuscript, authors used zeolites with the chabazite (CHA) structure type for computational studies, and proposed a unique point of view that both Al-O and Si-O bonds are susceptible to hydrolysis even at room temperature based on their calculation results. To confirm their assumption, the ^{17}O solid state NMR measurements were employed to monitor the migration of ^{17}O isotope in water into the framework of zeolite with different time scales, in particular, the key result of their ^{17}O analysis was the observation of ^{17}O signals from Al-O-Si and Si-O-Si linkages in the framework of zeolite after a short exposure to ^{17}O labeled water (ca. several minutes to 1 hour) . It is looks simply, but the experimental design is ingenious. However, the analysis at the initial stage of mixing with water should be more careful since authors emphasized the “fast” room temperature lability.

My main question also comes from the origination of the framework ^{17}O signals at the initial stage when authors used calcined sample mixed with H_2^{17}O . It is well-known that calcination and dehydration will result in partial breaking of Al-O as well as Si-O bond in zeolite framework, and thus the formation of framework defects is inevitable. As a result, the H_2^{17}O could easily react with framework defect to form new Al- ^{17}O -Si or Si- ^{17}O -Si structures in zeolite, whose contributions cannot be excluded at the initial stage of exposure. Thus, authors should provide more experimental evidence at the early stage of exposure, in which H_2^{17}O molecules induce new Si-O and Al-O bonds breaking under wet conditions at room temperature rather than react with the zeolite framework defect already formed after high-temperature calcination. I think this is of fundamental importance to prove “the fast room temperature lability”, since authors’ conclusion also conflicts with previous report by J. F. Stebbins et al. (SSNMR, 11, 1998, 243) that the framework oxygens did not exchange with H_2O in the channels at ambient temperature (within nine months), in which they used ^{17}O labeled zeolite sample and nature abundant H_2O . Nevertheless, after addressing the main question, I believe the author’s finding will be of interest for a broad readership within material science and chemistry communities. The work is publishable with the following additional concerns be addressed.

1. Why didn’t authors observed the ^{17}O signals from Bronsted acid sites (Si-OH-Al) that had been detected by C. P. Grey (Nature Materials, 4,2005,216) ? According to authors’ point of view, I think oxygen-16 on this site should be easily substituted by oxygen-17 in water at room temperature.
2. Authors proposed both Al-O and Si-O bonds are susceptible to hydrolysis even at room temperature based on their calculations. Could their calculations provide some dynamical

information, such as, how fast the hydrolysis or the oxygen substitutions occur when the water was introduced at room temperature?

3. The resolution of Fig. 1a was too poor to understand the proposed reaction mechanism, which should be improved.

Reviewer #2 (Remarks to the Author):

This paper reports an ab initio investigation of mechanisms through which Al-O and Si-O bonds in zeolites rapidly and reversibly break at 300K. This is confirmed through solid-state NMR measurements. The findings are very interesting and contradict the perception that zeolites are static entities. Advanced theoretical and experimental techniques have been used and given the novel insights that challenge common knowledge, I recommend publication in Nature Communications after addressing the following points.

- I agree with the authors that in the field of heterogeneous catalysis, zeolites are commonly seen as rigid and static materials. In the past, however, some (theoretical) studies have pointed out that this is not necessarily true. Some of that seminal work should be cited (e.g. the work of Smit and Keil DOI: 10.1021/jp0746446 or Treacy DOI: 10.1039/c003977b) as it clearly supports the current findings of the authors. I even suggest that the authors consider mentioning the similarity with the very flexible behavior in Metal-Organic Frameworks such as UiO-66, in which reversible bond breaking and forming at low temperature has been observed as well. This might even further emphasize the impact of the work.

- A brief discussion could be added on how the current findings potentially affect the nature of the active site and the catalytic behavior of the zeolite.

- Is it possible to quantify or estimate how many of these bonds simultaneously exhibit this behavior? To what extent is the observed lability dependent on the framework type?

- Recently, many studies appeared on zeolite dealumination and I suggest citing some of them due to very high relevance for the current work (e.g. Weckhuysen et al. DOI: 10.1021/acscatal.9b00307). Importantly, it has been discovered through DFT-based molecular dynamics simulations that under steaming conditions, water molecules cooperate during zeolite dealumination (Svelle et al. DOI: 10.1039/C9CY00624A).

- How was the initial loading of the zeolite chosen? It seems that the authors try to simulate a typical liquid water density in the zeolite, but it remains questionable to what extent such loading, even at low temperature, is reasonable or realistic? GCMC type calculations should be performed to

confirm that such loadings are in equilibrium with liquid water conditions outside the zeolite. How was the unit cell volume chosen? It is well-known that, especially in CHA type materials, the unit cell volume increases or decreases with increasing water loading in the pores.

- Panels a and b of Figure 1 are hardly readable. I recommend to improve the quality of that figure.

- A Si/Al ratio of 11 was chosen. How was the relative position of the Al atoms determined? How does this affect the results?

- It would be instructive to add clear figures of the chosen collective variables for the biased MD simulations.

- It is very unclear from which method free energies were calculated. In the methods section it is mentioned that metadynamics, slow growth and constrained MD simulations have been performed. It is, however, not further specified which simulation type was used for which results. It should also be mentioned how free energy barriers were calculated from the free energy profiles. Moreover, an estimation of error bars or convergence check of the free energy values is lacking.

- Unfortunately, no GIF type movies of the simulated mechanisms could be found in the SI.

- Behavior of water and the formation of protonated water clusters in CHA type zeolites has been extensively studied by a.o. the Van Speybroeck group. References to some of that work should be added.

- I suggest to present the numbers in Table 1 in a more appealing Figure/diagram (maybe combined with Figure 1?).

=====
Blue is a response to the author.

Green is changes in the manuscript
=====

Reviewer #1 (Remarks to the Author):

Probing the function of water in zeolite is an important and challenge issue, which is essential to understand stability and activity of zeolite. ^{17}O solid-state NMR coupling with ab initio calculations is a promising approach to explore how neutral liquid water at mild conditions affects the stability of the zeolite framework. In this manuscript, authors used zeolites with the chabazite (CHA) structure type for computational studies, and proposed a unique point of view that both Al-O and Si-O bonds are susceptible to hydrolysis even at room temperature based on their calculation results. To confirm their assumption, the ^{17}O solid state NMR measurements were employed to monitor the migration of ^{17}O isotope in water into the framework of zeolite with different time scales, in particular, the key result of their ^{17}O analysis was the observation of ^{17}O signals from Al-O-Si and Si-O-Si linkages in the framework of zeolite after a short exposure to ^{17}O labeled water (ca. several minutes to 1 hour) . It is looks simply, but the experimental design is ingenious. However, the analysis at the initial stage of mixing with water should be more careful since authors emphasized the “fast” room temperature lability.

My main question also comes from the origination of the framework ^{17}O signals at the initial stage when authors used calcined sample mixed with H_2^{17}O . It is well-known that calcination and dehydration will result in partial breaking of Al-O as well as Si-O bond in zeolite framework, and thus the formation of framework defects is inevitable. As a result, the H_2^{17}O could easily react with framework defect to form new Al- ^{17}O -Si or Si- ^{17}O -Si structures in zeolite, whose contributions cannot be excluded at the initial stage of exposure. Thus, authors should provide more experimental evidence at the early stage of exposure, in which H_2^{17}O molecules induce new Si-O and Al-O bonds breaking under wet conditions at room temperature rather than react with the zeolite framework defect already formed after high-temperature calcination.

The reviewer is, of course, correct that there are always defects present, albeit in small numbers, in the zeolites. In our ^{29}Si NMR we can identify the Q^3 silanol species (Si-OH) that are in the defect sites.

The incorporation of ^{17}O into Al-O-Si or Si-O-Si necessarily involves bond breaking and the formation of a (transient) defect, which can then heal through the mechanism shown. Therefore, it is impossible to separate whether the incorporation happens at an already preformed defect as opposed to one formed on exposure to moisture. As the reviewer suggests, one might expect that these may be the first sites to show exchange.

The experiments also show that the number of defects does not increase after exposure to ambient temperature liquid water. This means that the ^{17}O incorporation cannot be completely explained by reaction with defect sites, and that most of the incorporation must come from lability of the Si-O-Si and Al-O-Si units themselves. This is the important and surprising result that we report. Note that the initial zeolite is not dehydrated – therefore the defects have already been exposed to moisture in the

atmosphere: this removes the possibility of major changes in defect structure caused by rapid exposure to water.

Action: We have added a paragraph to page 10 to explain this more clearly.

I think this is of fundamental importance to prove “the fast room temperature lability”, since authors’ conclusion also conflicts with previous report by J. F. Stebbins et al. (SSNMR, 11, 1998, 243) that the framework oxygens did not exchange with H₂O in the channels at ambient temperature (within nine months), in which they used ¹⁷O labeled zeolite sample and nature abundant H₂O. Nevertheless, after addressing the main question, I believe the author’s finding will be of interest for a broad readership within material science and chemistry communities. The work is publishable with the following additional concerns be addressed.

Our work is not at all at odds with the Stebbins paper. In the Stebbins paper they enriched using water at 200C and 40 bar for 1 month. They also back react at 196C and 23 Torr for 3 days. Therefore, this is very much more akin to steaming of zeolites (which we discuss in the introduction). They do comment however that on exposing the enriched sample to (wet) air they see no back exchange after 9 months. We would agree with this, we see little change in samples we have enriched after many months simply from air. There is an important and well-known difference between liquid and vapour phase water in its reactivity (at all temperatures) – see the many papers discussing reactions with ‘hot liquid water’. The reaction we demonstrate is in liquid water at room temperature. If we leave the zeolites only in contact with wet air, as Stebbins did, we see no change - exactly the same as Stebbins.

Action: We have added a paragraph to page 10 to explain this.

1. Why didn’t authors observed the ¹⁷O signals from Bronsted acid sites (Si-OH-Al) that had been detected by C. P. Grey (Nature Materials, 4,2005,216) ? According to authors’ point of view, I think oxygen-16 on this site should be easily substituted by oxygen-17 in water at room temperature.

This Grey paper concerns zeolite Y and there is a similar one on ¹⁷O of mordenite (J Am Chem Soc 2012, 134, 9708-9720). In both cases the sample is dehydrated and so the Bronsted acid proton is attached to the framework. However, it has been known for more than 20 years (from neutron diffraction, solid-state NMR and computer simulations) that in hydrated zeolites the equilibrium (average) position is that the Bronsted acid proton is removed from the framework and is solvated by the water. Therefore, in hydrated zeolites you will never see the Si-OH-Al unit by NMR. A good reference to start with is part of the extensive recent work by Lercher (DOI: 10.1021/acs.chemmater.7b02133).

Action: We have not added to the paper on this point as it is well known. If required we could add the dehydrated and hydrated ²⁷Al and ¹H spectra of both hydrated and dehydrated materials, which shows this phenomenon very well.

2. Authors proposed both Al-O and Si-O bonds are susceptible to hydrolysis even at room temperature based on their calculations. Could their calculations provide some dynamical information, such as, how fast the hydrolysis or the oxygen substitutions occur when the water was introduced at room temperature?

In principle, it is possible to understand the kinetics of the initial hydrolysis (using the Eyring equation, for example). For Si-O and Al-O bond hydrolysis, applying the equation using the calculated values of

free energy barriers (Table S2) gives rise to approximate rate constants of 6×10^1 and 7×10^6 (7×10^8) for Si-O₁/O₄ and Al-O₁/(O₄) bonds, respectively (rate constant unit depends on the order of reaction, e.g., s⁻¹ for first order and M.s⁻¹ here assumed second order). Interpreting the rate constant as a number of reactive events per reactant unit (bond, molecule, etc.) and per unit of time, [10.1002/anie.196904381] the hydrolysis of both Si-O and Al-O bonds should be extremely rapid at RT with tens (Si-O) to millions (Al-O) of bond breaking events per second and per reactant unit. However, the calculations should be taken with caution, and thus we do not present them in the manuscript. Rate and equilibrium constants are **extremely** sensitive to the precision of the computed data, e.g., an error of 5 kJ/mol translates into an order of magnitude change in reaction/equilibrium constants. The statistical analysis of our data indeed shows that standard deviations of our reaction free energies and free energy barriers are approx. 5 kJ/mol. In addition, there is the error introduced by the choice of DFT functional, which is likely to be of a similar or slightly larger size. Nevertheless, our estimates agree qualitatively with what is observed in experiment, i.e., a fast ¹⁷O exchange in Al/Si-O-Si species within tens of minutes after combination of enriched water with zeolite.

Action: We feel that the strong dependency of the rate constants and other dynamic information on the calculations precludes us from being too strong in reporting the numbers from these calculations. We could put them in the SI but feel they could mislead some readers so we prefer it if we do not emphasise them too much.

3. The resolution of Fig. 1a was too poor to understand the proposed reaction mechanism, which should be improved.

Action: We have improved and amended figure 1 to more clearly describe the reaction mechanisms and energetics.

Reviewer #2 (Remarks to the Author):

This paper reports an ab initio investigation of mechanisms through which Al-O and Si-O bonds in zeolites rapidly and reversibly break at 300K. This is confirmed through solid-state NMR measurements. The findings are very interesting and contradict the perception that zeolites are static entities. Advanced theoretical and experimental techniques have been used and given the novel insights that challenge common knowledge, I recommend publication in Nature Communications after addressing the following points.

- I agree with the authors that in the field of heterogeneous catalysis, zeolites are commonly seen as rigid and static materials. In the past, however, some (theoretical) studies have pointed out that this is not necessarily true. Some of that seminal work should be cited (e.g. the work of Smit and Keil DOI: 10.1021/jp0746446 or Treacy DOI: 10.1039/c003977b) as it clearly supports the current findings of the authors. I even suggest that the authors consider mentioning the similarity with the very flexible behavior in Metal-Organic Frameworks such as UiO-66, in which reversible bond breaking and forming at low temperature has been observed as well. This might even further emphasize the impact of the work.

We are very happy to add a section to discuss the references noted. However, it should be understood that the type of flexibility proposed for zeolites does not involve bond breaking – this has been known for a while and we might think of this as a physical flexibility – it does not involve lability of the bonds and so is fundamentally different to the work we report. Similarly, in MOF chemistry there has been a focus on similar physical flexibility (often called ‘breathing’). However, there is more of a realization in MOFs that there is lability of bonds possible – often this is called hemilability in MOFs when it doesn’t lead to structural degradation.

Action: We have added a section to discuss the different types of flexibility possible in zeolites (and MOFs) and add the requested references plus a review article on lability in MOFs on page 10.

- A brief discussion could be added on how the current findings potentially affect the nature of the active site and the catalytic behavior of the zeolite.

While the process we report happens at room temperature, well away from usual catalysis conditions, we do agree that this is an important point.

Action: We have added a small section to discuss the likely effects of our findings in terms of catalytically relevant properties (pore size/diffusion). This has been included on page 6: There is also a short addition to the discussion/conclusions section with re-emphasises the importance of the findings (page 11).

- Is it possible to quantify or estimate how many of these bonds simultaneously exhibit this behavior? To what extent is the observed lability dependent on the framework type?

As we show in the manuscript lability depends on the local topology within a framework, as evidenced by the different results for non-equivalent oxygens within the same zeolite (Figure 1). The primary underlying cause of these variations is likely to be accessibility, as in the case of Si-O-Si hydrolysis, this dictates whether the low-barrier collective “axial” mechanism is possible. Therefore, we would, with certainty, expect that lability will be framework-type dependent as almost by definition the local topology will differ between different frameworks.

As for the simultaneous bond hydrolysis, a simple back-of-the-envelope calculation gives an estimate of 8% of Si-O-Si bonds to be hydrolyzed simultaneously (dominantly O4 bonds); However, the proviso regarding the accuracy of the computational studies and these figures also holds here (see above and below).

- Recently, many studies appeared on zeolite dealumination and I suggest citing some of them due to very high relevance for the current work (e.g. Weckhuysen et al. DOI: 10.1021/acscatal.9b00307). Importantly, it has been discovered through DFT-based molecular dynamics simulations that under steaming conditions, water molecules cooperate during zeolite dealumination (Svelle et al. DOI: 10.1039/C9CY00624A).

Dealumination has received a great deal of attention recently. Both of the suggested papers investigate the dealumination process under steaming conditions, one of them appeared only after we submitted our paper. Both papers are focusing on the cooperativity between the first and the second water involved in the reaction, while we are investigating fully hydrated zeolite channels at ambient conditions. We agree with the referee that both references should appear in our manuscript.

Action: We have added the appropriate references on pages 2 and 3.

- How was the initial loading of the zeolite chosen? It seems that the authors try to simulate a typical liquid water density in the zeolite, but it remains questionable to what extent such loading, even at low temperature, is reasonable or realistic? GCMC type calculations should be performed to confirm that such loadings are in equilibrium with liquid water conditions outside the zeolite.

The water loading was chosen according to the maximum loading of a fully hydrated system as described in *Simulation Model* subsection of *Methods* section. The water loading of 5 water molecules per chabazite cage agrees well with the experimental estimate of 4-5 water molecules in micropore volume from the water adsorption isotherm of an H-CHA sample (Si/Al=11) at 293 K (see Figure S4), i.e., a sample very similar to our alumino-silicate CHA model. The micropore-filling capacity was estimated from the Dubinin equation [[10.1016/0144-2449\(89\)90004-3](https://doi.org/10.1016/0144-2449(89)90004-3)]. Even higher micropore-filling capacities (5-7 water molecules) are reported for natural chabazites (Ca/Na exchanged) with lower Si/Al ratios at similar temperatures [[10.1016/0144-2449\(89\)90004-3](https://doi.org/10.1016/0144-2449(89)90004-3), [10.1016/S0167-2991\(08\)80271-6](https://doi.org/10.1016/S0167-2991(08)80271-6), [10.1016/0022-3697\(61\)90206-2](https://doi.org/10.1016/0022-3697(61)90206-2)]. In addition, the same scheme to determine the number of water molecules which gives a density of 1g/ml in the free pore volume was previously applied to another hydrated zeolite system (UTL), for which the model value (18 molecules/38 T atom unit cell) agreed well with the experimental value (20).

Action: We have added a section to the supporting information containing the experimental water adsorption isotherm (S4). We have also introduced the experimental corroboration of our model to the methods section of the manuscript (page 12).

How was the unit cell volume chosen? It is well-known that, especially in CHA type materials, the unit cell volume increases or decreases with increasing water loading in the pores.

We thank the reviewer for the valuable comment regarding the flexibility of CHA type materials upon water loading. It has been nicely shown by others that the cell dimensions can vary in a complex, non-monotonic manner as a function of water loading and temperature [[10.1021/acscatal.5b02139](https://doi.org/10.1021/acscatal.5b02139), [10.1039/C9CY00624A](https://doi.org/10.1039/C9CY00624A)]. It is certainly possible that the cell volume will vary between reactant and product states of hydrolysis, between the two models (Si-only CHA and Si/Al=11 H-CHA), and with Al distribution. However, we do not expect that the flexibility of CHA unit cell will have a significant effect on the mechanism of hydrolysis or on the reported lability of Si-O-Al and Si-O-Si bonds under high water loading conditions. In our calculations, the CHA unit cell was taken from the IZA zeolite database, locally re-optimized in the presence of 15 H₂O molecules at Si/Al=11 and then kept fixed during all simulations.

Action: The selection of CHA structure was added to the methods section of the manuscript on page 12.

- Panels a and b of Figure 1 are hardly readable. I recommend to improve the quality of that figure.

Action: We have improved and amended figure 1 to more clearly describe the reaction mechanisms and energetics. We have also taken into account the suggestion to incorporate the energetics from table 1 into figure.

- A Si/Al ratio of 11 was chosen. How was the relative position of the Al atoms determined? How does this affect the results?

A distribution of Al atoms was chosen so that each Al is separated from each other by three silicon atoms (i.e. eight bonds). This ensures the coupling between aluminium atoms is minimal. The solvation of protons from the framework Brønsted acid sites is facile, which means that differences between protons initially located at inequivalent sites are lost upon equilibration (solvation). We did not consider lower Si:Al ratios than 11, as this choice is consistent with the experiment. However, at low ratios, it is likely that the Al distribution will become important. In recent work [Chem. Sci. 2019, **10**, 5705-5711] we considered the effect of Loewenstein-rule-disobeying (Al-O(H)-Al) groups on the stability of Al distributions in CHA. These moieties reduce proton acidity and may affect the reactivity. However, given the propensity of zeolites to obey the Loewenstein rule, and the relatively high Si:Al ratio in the current work, we expect the choice of Al distribution to have little effect on the results, provided Al atoms are sufficiently far apart.

- It would be instructive to add clear figures of the chosen collective variables for the biased MD simulations.

Action: We have now added a figure to the Supporting Information to this effect (figure S5).

- It is very unclear from which method free energies were calculated. In the methods section it is mentioned that metadynamics, slow growth and constrained MD simulations have been performed. It is, however, not further specified which simulation type was used for which results. It should also be mentioned how free energy barriers were calculated from the free energy profiles.

The description of the theoretical methods has been clarified in the methods section, including mention of the calculation of free energy barriers through thermodynamic integration of free energy gradients obtained from blue moon sampling (constrained dynamics along CV path). The MTD has been explained to have been used as an exploratory method to determine good CVs, rather than as a means to generate accurate free energy estimates, as is sometimes the case in other works. The appropriate reference to methodology has been included in the main text to remove ambiguity.

Action: We have added a considerable section to the methods section.

Moreover, an estimation of error bars or convergence check of the free energy values is lacking.

We have included an error analysis to estimate the precision of the free energies determined via constrained MD. The errors are found to be around +/-5 kJ/mol for all systems considered.

Action: The error analysis has been added to the free energy plots in the supporting information (Figure S5), along with a description of the method by which the precision was estimated.

- Unfortunately, no GIF type movies of the simulated mechanisms could be found in the SI.

This is our mistake – we forgot to amend the manuscript after we realized that GIFs were not supported by Nature Communications. We will upload the movies in a better format for the revised article.

- Behavior of water and the formation of protonated water clusters in CHA type zeolites has been extensively studied by a.o. the Van Speybroeck group. References to some of that work should be added.

This is true. There are relevant papers which consider proton mobility and water-BA proton interactions with advanced theoretical techniques. We appreciate the suggestion and have added some important references to page 3.

Action: We have added the following section to the manuscript “The CHA framework in the form of SAPO-34 has been interrogated with molecular dynamics simulations for applications in methanol conversion, where the interaction of incorporated water and reactant molecules is important.^{26, 27} “

- I suggest to present the numbers in Table 1 in a more appealing Figure/diagram (maybe combined with Figure 1?).

We have taken this good advice and amended figure one accordingly.

Action: We have produced a new Figure 1 which should hopefully satisfy the reviewer

Reviewer #3:

1. The main topic of this contribution is the lability of Al-O and Si-O at room temperature. For synthesis and dealumination of zeolites the reactivity of Al-O and of Si-O are of critical importance. However, the mechanisms of bond braking and making are hardly studied. The situation is even more contrasted at room temperature for which the belief is no reactivity. Though such a situation has been demonstrated not to be the case even with C-H bond of alkanes at room temperature on zeolites. Furthermore, aluminum zoning can also be understood if bond braking and making would be clearly understood. It is well know that the zoning keeps evolving at room temperature.

This contribution clarifies therefore a phenomenon with many consequences for zeolites processes.

If qualitatively the stability of Al-O bond is lower that Si-O, the lability of Al-O is higher than Si-O, some sort of seemingly paradoxical situation explained by the much easier high coordination states of aluminum atom (5 and 6) than for silicon atom.

However, no mechanism that would bring Si-O lability close to Al-O has been considered up to now. Therefore, the mechanism proposed by the authors allows to take into account the experimental data of aluminum redistribution in a zeolite at room temperature, with time.

2. I consider therefore this study as of a high scientific information level, and would find appropriate to accept it for publication in your journal.

3. I noticed a small but annoying problem with Figure 2. The scales of 1D spectra and of the direct dimension of the 3QMAS are not the same, leading to a non optimum readability of the spectral data. I would recommend to fix it.

Action: We have replaced figure 2 with one where the scales match.

The authors could quote the seminal work of Kemplerer for using ¹⁷O as a way to analyze bond lability in titanium complexes.

Action: These are good papers to support our choice of experiment and so we have added one of Klemperer's reviews in the area

REVIEWERS' COMMENTS:

Reviewer #1 (Remarks to the Author):

I am satisfied with the responses and the modifications made by the authors, and recommend the revised manuscript published in Nature Communication as is

Reviewer #2 (Remarks to the Author):

The authors thoroughly revised the manuscript and properly addressed my concerns. The manuscript is now ready for publication.